Nonlinear relationship between circulating natural killer cell count and 1-year relapse rates in myasthenia gravis: a retrospective cohort study

Zhang Yifan
Wen Zhiguo
Chen Meiqiu
Xia Cong
Cai Fang
Chu Lan chulan8899@sina.com
Department of Neurology, The Affiliated Hospital of Guizhou Medical University , Guiyang , Guizhou , China
Jurisic Vladimir
Electronic publication date: 2024 Dec 6
Publication date: 2024
Volume: 12
Electronic Location ID: e18562
Received 2024 Jan 25; Accepted 2024 Oct 31
Copyright: ©2024 Zhang et al.
Copyright year: 2024
Copyright holder: Zhang et al.
License: This is an open access article distributed under the terms of the Creative Commons Attribution License, which permits unrestricted use, distribution, reproduction and adaptation in any medium and for any purpose provided that it is properly attributed. For attribution, the original author(s), title, publication source (PeerJ) and either DOI or URL of the article must be cited.
License URL: https://creativecommons.org/licenses/by/4.0/

Keywords: Myasthenia gravis, Natural killer cells, Recurrence, Nonlinear

Funding: the Guiyang Science and Technology Bureau [2022]-4-2-8 This study was funded by the Guiyang Science and Technology Bureau. (NO. [2022]-4-2-8). The funders had no role in study design, data collection and analysis, decision to publish, or preparation of the manuscript.

==============================
Background

The relapse rate in myasthenia gravis (MG) is high, and promising therapies have emerged; however, identifying potential predictive factors for relapse remains a challenge. This study aimed to explore the association between circulating natural killer (NK) cell levels and the risk of recurrence in MG.

Methods

This retrospective cohort study included 265 patients with MG whose data were included in the Neurology Department of the Affiliated Hospital of Guizhou Medical University database between March 2015 and March 2022. Data from electronic medical records were collected, which included the patients’ circulating NK cell count (exposure variable) and demographic/clinical characteristics (covariates). The primary outcome was the 1-year MG recurrence rate.

Results

The study revealed a non-linear relationship between peripheral NK cell count and MG recurrence, with an inflection point at 5.38. Below this threshold, the risk of recurrence was low with higher NK cell counts (relative risk (RR): 0.23, 95% confidence interval (CI) [0.11–0.490]); above this threshold, no significant association was observed (RR: 1.43, 95% CI [0.62–3.34]). Furthermore, the NK cell proportion showed no significant linear or non-linear association with MG recurrence risk (RR: 0.84, 95% CI [0.57–1.2]).

Conclusion

This study provides epidemiological evidence of a potential association between peripheral NK cell count and MG recurrence risk, suggesting an immunoregulatory protective effect within a specific NK cell count range. These findings may inform more personalized MG treatment strategies, warranting further validation in larger and more diverse cohorts.

Introduction

Myasthenia gravis (MG) is an autoimmune disorder characterized by the dysfunction of neuromuscular transmission, leading to localized fatigue and weakness, especially in the muscles around the eyes and proximal limbs (Berrih-Aknin & Le Panse, 2014; Pevzner et al., 2012). The age-adjusted incidence of MG is 5–30 cases per 100,000 individuals, with a slightly higher incidence in women than in men (Green et al., 2020; Niks, Kuks & Verschuuren, 2007; Dresser et al., 2021). The outcome of MG is significantly associated with disease severity, potential complications, and treatment effectiveness (Su et al., 2022a; Su et al., 2022b). Risk factors associated with an unfavorable prognosis in patients include advanced age at onset, thymoma, and elevated levels of acetylcholine receptor antibodies (Gilhus, 2017; Rodolico et al., 2021; Nelke et al., 2022). Although promising new therapies have emerged in clinical practice, acetylcholinesterase inhibitors and glucocorticoids remain primary treatments for MG (Arroyo, 2019; Gilhus et al., 2016). Nevertheless, the relapse rate ranges from 18 to 34%, and only 10–20% of patients attain complete stable remission after the cessation of immunotherapy (Su et al., 2022a; Su et al., 2022b).

Natural killer (NK) cells are crucial innate lymphocytes that are involved in immune regulation via cell lysis and cytokine secretion (Clark et al., 2019). NK cells do not express antigen-specific receptors but can rapidly respond to stressed, transformed, or infected cells. They are classified into different subsets based on the expression of surface markers such as CD56 and CD16. NK cells play a crucial role in the innate immune response through their cytotoxic activity and cytokine production (Dzopalić, Božić-Nedeljković & Jurišić, 2019). Irregularities in the count and function of NK cells have been associated with the onset and progression of various diseases, including systemic lupus erythematosus and rheumatoid arthritis (Ciafaloni, 2019; Park et al., 2009; Segerberg et al., 2019; McInnes & Schett, 2011). Moreover, several clinical studies and animal experiments have found a significant association between MG and NK phenotype and count (Ge et al., 2023; Liu et al., 2021; Yang et al., 2019; Chien et al., 2011; Nguyen et al., 2006). These results suggest that NK cells play a role in the development of MG; however, epidemiological studies are needed to elucidate this relationship.

Given the evidence regarding the involvement of NK cells in MG pathogenesis and disease activity, we hypothesized that NK cells influence the prognosis of patients with MG. We aimed to describe the relationship between the two factors and offer further evidence regarding the association between NK cell counts and MG progression.

Materials & Methods

Study population

A clinical database was retrospectively searched to obtain data of patients treated at the Neurology Department of the Affiliated Hospital of Guizhou Medical University, Guizhou Province, China. The data collection period was from March 2015 to March 2022. Patient information was collected continuously and nonselectively by extracting data from the hospital’s electronic medical record system. We enrolled patients who met the following inclusion criteria: (1) aged 18 years or older; (2) exhibiting clinical symptoms and signs consistent with MG; and (3) diagnosed with MG based on nerve conduction studies, acetylcholine receptor antibody detection, and other relevant tests. The sample comprised 354 eligible patients, and after excluding potential interference, the final dataset included 265 patients (Fig. 1). The exclusion criteria were as follows: (1) not meeting the diagnostic criteria for MG; (2) lack of information about NK cell levels in peripheral blood samples or about follow-up data; and (3) other severe chronic diseases, tumors, systemic autoimmune diseasies, or active infectons. This study was approved by the Ethics Committee of the Affiliated Hospital of Guizhou Medical University, [No.(2022)513]. As this was a retrospective study, patients’ information was de-identified and informed consent was not required.

Figure 1 Flowchart of patient selection.

Outcome variable

The study outcome variable was the rate of MG recurrence within 1 year after treatment, which was represented as a binary variable (Y = 0 for no recurrence, Y = 1 for recurrence). The recurrence criteria were based on the following indicators: symptom recovery and return or worsening of typical MG symptoms, including muscle weakness, fatigue, ptosis, and diplopia. Recurrence was considered if the symptoms reappeared or worsened. Further, muscle strength assessment involved evaluating the patients’ muscle strength, particularly in previously weak muscle groups, where significant weakness or asymmetry may indicate MG recurrence. Serum acetylcholine receptor antibody testing was considered crucial, and retesting serum antibody levels during recurrence supported the diagnosis if the antibodies reappeared or their levels increased. Monitoring the response to common MG treatments and observing the worsening of symptoms after dose reduction or discontinuation also suggested MG recurrence. Electrophysiological examinations, including repetitive nerve stimulation and single-fiber electromyography, were used to assess nerve muscle transmission function. Abnormal results were considered indicative of recurrence, possibly including nerve conduction block or fibrillation. Physicians were unaware of the patients’ baseline NK cell values while assessing MG recurrence.

Exposure variable

The exposure variable was the number of NK cells in peripheral blood samples, which included both the relative proportion and the absolute NK cell count. NK cells were defined as cells expressing CD56 and not expressing CD3 surface markers (CD3- CD56+). All patients underwent baseline flow cytometry testing of the peripheral blood before the initiation of treatment. Specifically, blood samples with ethylenediaminetetraacetic acid (EDTA) were collected from peripheral blood and analyzed using a BD FACSCalibur flow cytometer (BD Biosciences, Franklin Lakes, NJ, USA). Peripheral blood samples were collected using EDTA-K2 as an anticoagulant. A reverse pipetting technique was employed to transfer 50 µL of well-mixed anticoagulated blood to the bottom of a tube containing absolute counting beads. Subsequently, 20 µL of a six-color antibody cocktail (CD3-FITC/CD16+56-PE/CD45-PerCP-Cy5.5/CD4-PC7/CD19-APC/CD8-APC-Cy7; Beijing Tongsheng Times Biotechnology Co., Ltd., Beijing, China) was added (Table S1), and the mixture was incubated at room temperature in the dark for 25 min. Subsequently, 450 µL of flow cytometry lysing solution (BD Biosciences, USA) was added to the sample, which was then thoroughly mixed and incubated at room temperature in the absence of light for an additional 10 min before analysis on a FACS-Canto II flow cytometer (BD Biosciences, Franklin Lakes, NJ, USA). A minimum of 2,500 events were recorded over a period of 10 min, and data analysis was conducted using FACSDiva software. Gating strategy: First, singlets were selected and double peaks were excluded in the FSC-A/FSC-H dot plot. Fragments and dead cells were removed and the lymphocyte population was first identified in the FSC-A/SSC-A dot plot using forward and side scatter characteristics. Next, CD45+CD3+ T cells and CD45+CD3- non-T cells are distinguished in the CD45/CD3 dot plot using linear gating. Finally, double positive labelling for CD16 and CD56 is used in the CD16/CD56 dot plot and quadrant gating is applied within the CD3- lymphocyte population to finally identify NK cells (Fig. 2). All flow cytometry procedures were performed according to standardized protocols provided by the manufacturer All tests were performed in the central laboratory of the Affiliated Hospital of Guizhou Medical University, with the laboratory staff blinded to the clinical information of the samples. In addition, the researchers were unaware that these results could be used for future data analyses.

Figure 2 Detailed Gating Strategy for NK Cell Identification.

Covariates

The covariates in this study included demographic characteristics (sex and age), MG-related variables (thymus status, Osserman classification, involved muscle groups, expression status of muscle-specific kinase antibodies (MuSK-Ab), and acetylcholine receptor antibodies (AchR-Ab)), and treatment strategies (cholinesterase inhibitors, glucocorticoids, immunosuppressants, intravenous immunoglobulin, plasmapheresis, rituximab, and thymectomy). All covariates were measured at baseline. These covariates were selected based on their potential association with relapse, as indicated by previous studies and clinical experience.

Statistical analyses

Descriptive statistics were used to summarize the data—continuous normally distributed variables are presented as mean ± standard deviation, while variables with other distributions are presented as median (range). Categorical variables are expressed as proportions. After transforming the NK cell count using a natural logarithm (Ln) and categorizing it into quartiles, the distributions of other covariates and outcome variables across the defined groups were examined. Among-group comparisons of means were conducted using one-way analysis of variance for normally distributed variables and the Kruskal–Wallis rank sum test for variables with other distributions. Categorical variables were compared using the chi-square test.

Our primary aim was to investigate the true association between exposure and outcome variables while controlling for confounding factors. Initially, a multivariable logistic regression model was used to evaluate the independent effects of each exposure variable (NK cell levels) on MG recurrence, adjusting for confounding factors, and the results were presented as effect estimates with 95% confidence intervals (CIs). Covariate selection was performed prior to adjustment; the P-value for the covariate association with MG recurrence was <0.1 (Kuhn et al., 2010). Considering the limitations of binary logistic regression in handling nonlinear relationships, we used generalized additive models to address this issue. Initially, penalized spline methods were used to smooth the relationship between NK cell levels and MG recurrence. Subsequently, the inflection points in the nonlinear relationship were identified using a recursive algorithm, and piecewise linear models were constructed to elucidate the nonlinear relationship at these points.

Sensitivity analysis was performed to confirm the reliability of the results by using models with varying levels of adjustment. The stability of the results was assessed by examining the changes in the effect estimates across models. Furthermore, the exposure variable was categorized into three groups (quartiles), and trend tests were conducted to observe the consistency of trends between the categorical and continuous variables.

The missing data accounted for less than 2% of the dataset, and multiple imputation methods were not employed (Table S2). All the analyses were conducted using R software (version 3.6.1, R Core Team (2020)) and EmpowerStats (X&Y Solutions, Inc., Boston, MA, USA, http://www.empowerstats.com). Statistical significance was defined by a two-tailed P-value of <0.05.

Table 1 Baseline characteristics of patients with MG.

Variables	Ln (Natural Killer cell count)	P-value	
Sociodemographic variables			
Age, n (%)		0.796	
≤median (46 years old)	5.31 ± 0.74		
>median (46 years old)	5.34 ± 0.76		
Sex, n (%)		0.400	
Male	5.39 ± 0.68		
Female	5.30 ± 0.78		
MG-related variables			
Thymus status, n (%)		0.181	
Normal	5.38 ± 0.75		
Abnormal	5.26 ± 0.74		
Osserman classification, n (%)		0.018	
I	5.41 ± 0.70		
II	5.34 ± 0.79		
III	4.99 ± 0.70		
respiratory muscles, n (%)		0.236	
Not involved	5.34 ± 0.75		
Involved	5.13 ± 0.69		
Muscles of the limbs, n (%)		0.110	
Not involved	5.39 ± 0.74		
Involved	5.24 ± 0.76		
Muscles of the pharyngeal muscles, n (%)		0.048	
Not involved	5.39 ± 0.73		
Involved	5.20 ± 0.77		
Muscles of the eyes and extraocular muscles, n (%)		0.395	
Not involved	5.21 ± 0.67		
Involved	5.34 ± 0.76		
AChR-ab, n (%)		0.273	
Negative	5.39 ± 0.70		
Positive	5.29 ± 0.77		
MUSK-ab, n (%)		0.380	
Negative	5.32 ± 0.74		
Positive	5.62 ± 0.92		
Treatment strategies			
Thymectomy , n (%)		0.539	
No	5.31 ± 0.72		
Yes	5.38 ± 0.84		
Acetylcholinesterase inhibitors, n (%)		0.015	
No	5.16 ± 0.74		
Yes	5.40 ± 0.74		
Corticosteroids, n (%)		0.939	
No	5.34 ± 0.72		
Yes	5.33 ± 0.76		
Immunosuppressants, n (%)		0.398	
No	5.31 ± 0.72		
Yes	5.40 ± 0.83		
Intravenous immunoglobulin, n (%)		0.248	
No	5.35 ± 0.76		
Yes	5.21 ± 0.69		
Plasma exchange, n (%)		0.030	
No	5.33 ± 0.75		
Yes	4.99 ± 0.36		
Rituximab, n (%)		0.919	
No	5.33 ± 0.73		
Yes	5.31 ± 0.93		

Results

Baseline characteristics of participants

Table 1 shows the baseline characteristics of patients with MG. Age, sex, thymus status, Osserman classification, respiratory muscle/limb/pharyngeal muscle involvement, ocular muscle involvement, and AChR-Ab and MuSK-Ab status were not significantly associated with NK cell counts. However, patients with Osserman class III had significantly lower NK cell counts than those of patients with class I (P = 0.018). Additionally, patients with pharyngeal muscle involvement had significantly lower NK cell counts than those of patients without (P = 0.048). Patients using acetylcholinesterase inhibitors had significantly higher NK cell counts than those of patients not using these drugs (P = 0.015), while patients receiving plasma exchange had significantly lower NK cell counts than those of patients not receiving plasma exchange (P = 0.030). Other treatment strategies were not significantly associated with NK cell counts.

Univariate and multivariate analyses findings

We investigated the relationship between Ln-transformed peripheral NK cell count (Ln(PNKc)) and 1-year relapse risk in patients with MG using univariate and multivariate logistic regression models (Table 2). In the unadjusted model, the results demonstrated a 53% (95% CI [0.32–0.69]) decrease in the 1-year relapse risk for each unit increase in Ln(PNKc). After adjusting for sex and age, the results remained consistent. Upon adjusting for the preselected covariates (Tables S3 and S4), each unit increase in Ln-transformed PNKc resulted in a 51% decrease in the risk of relapse (95% CI [0.32–0.75]). For the sensitivity analysis, we categorized Ln(PNKc) into three equal groups and calculated the trend P-value. After adjustments for age, thymus status, thymectomy, AChR-Ab, MuSK-Ab, Osserman classification, involvement of pharyngeal muscles, acetylcholinesterase inhibitor use, and immunosuppressant use, we established a significant negative association between Ln(PNKc) and 1-year relapse risk in patients with MG.

Table 2 Univariate and multivariable model results on NK cell count.

Exposure	Non-adjusted
RR, 95%CI, P values	Adjust I
RR, 95%CI, P values	Adjust II
RR, 95%CI, P values	
Ln (Peripheral blood NK cell count)	0.47 (0.32, 0.69) <0.001	0.45 (0.31, 0.67) <0.0001	0.49 (0.32, 0.75) 0.001	
Peripheral blood NK cell count (Tertile)				
Low	Reference	Reference	Reference	
Middle	0.33 (0.17, 0.62) 0.001	0.32 (0.17, 0.62) 0.001	0.35 (0.17, 0.72) 0.004	
High	0.39 (0.21, 0.72) 0.0030	0.38 (0.20, 0.72) 0.0032	0.47 (0.23, 0.96) 0.038	
P for trend	0.60 (0.44, 0.84) 0.0023	0.60 (0.43, 0.84) 0.0026	0.67 (0.46, 0.96) 0.029	
Notes.

RR risk ratio

CI confidence interval

NK nature killer

Non-adjusted model adjust for: None

Adjust I model adjust for: Gender; Age

Adjust II model adjust for: Age; thymus status; Thymectomy; AChR-ab; MUSK-ab; Osserman classification; involvement of pharyngeal muscles; acetylcholinesterase inhibitor use; Immunosuppressants use

We also evaluated the relationship between the proportion of NK cells and the risk of 1-year recurrence in patients with MG (Table 3). The results revealed that for both unadjusted and demographically adjusted models, a single unit increase in Ln(NK cell proportion) corresponded to a 35% and 32% reduction in the 1-year recurrence risk for patients with MG, respectively; however, the P-values were not statistically significant. Following adjustment for preselected covariates (Table S5), no significant association was observed between Ln(NK cell proportion) and the risk of recurrence in patients with MG (odds ratio [OR]: 0.84, 95% CI [0.57–1.25], P = 0.3898). The sensitivity analysis demonstrated that using Ln(NK cell proportion) as a categorical variable (terciles) yielded a trend P-value that was consistent with its use as a continuous variable.

Table 3 Univariate and multivariable model results on NK cell proportion.

Exposure	Non-adjusted
RR, 95%CI, P values	Adjust I
RR, 95%CI, P values	Adjust II
RR, 95%CI, P values	
Ln (NK cells proportion)	0.65 (0.46, 0.94) 0.0205	0.68 (0.47, 0.99) 0.0452	0.84 (0.57, 1.25) 0.3898	
Ln (NK cells proportion) (tertile)				
Low	Reference	Reference	Reference	
Middle	0.61 (0.34, 1.10) 0.1022	0.67 (0.37, 1.22) 0.1890	0.77 (0.40, 1.47) 0.4224	
High	0.66 (0.37, 1.19) 0.1686	0.72 (0.40, 1.32) 0.2932	1.00 (0.52, 1.93) 0.9997	
P for trend	0.81 (0.60, 1.09) 0.1625	0.85 (0.63, 1.15) 0.2893	1.00 (0.71, 1.39) 0.9787	
Notes.

Non-adjusted model adjust for: None

Adjust I model adjust for: gender; age

Adjust II model adjust for: Age; thymus status; Thymectomy; AChR-ab; MUSK-ab; Osserman classification; involvement of pharyngeal muscles; acetylcholinesterase inhibitor use; Immunosuppressants use

Nonlinearity

We explored the potential non-linear association between Ln(PNKc) and the proportion and risk of recurrence in MG. Smoothed curve fitting revealed a saturation effect in the non-linear relationship between Ln(PNKc) and the risk of MG recurrence. Conversely, the relationship between Ln(PNKc ratio) and risk of MG recurrence was linear. We used the P-values of the log-likelihood ratio for comparison, which indicated that the two-piecewise linear model fit the relationship between Ln(PNKc) and the risk of MG recurrence better than did the logistic regression model. Using recursive algorithm calculations, we determined the inflection point to be 5.38. Higher levels of Ln(PNKc) were associated with a lower risk of MG recurrence when Ln(PNKc) was within the range of 3.00–5.38. The model-based relative risk (RR) can be interpreted as a 77% reduction in the 1-year recurrence risk for patients with MG per unit increase in Ln(PNKc) (RR: 0.23, 95% CI [0.1–0.49]). However, within the range of 5.38–7.56, we could not observe any significant association between Ln(PNKc) and the risk of MG recurrence (RR: 1.24, 95% CI [0.57–2.70]).

The logistic regression model yielded an OR of 0.84 (95% CI [0.57–1.25]; P = 0.3898). This result indicated that there was no statistically significant association between Ln(NK cell proportion) and 1-year relapse risk in patients with MG using the logistic regression model. An inflection point of 1.81 was identified upon applying the two-piecewise linear model. Below the inflection point, the effect size was estimated at 0.28 (95% CI [0.06–1.24], P = 0.0935). Above the inflection point, the effect size was estimated at 1.16 (95% CI [0.65–2.05], P = 0.6203). These findings indicated that a trend towards a protective effect existed below the inflection point, although it was not statistically significant. However, above the inflection point, no significant association was observed. The P-value for the log-likelihood ratio test was 0.114, indicating that the two-piecewise linear model did not significantly improve the fit compared to the logistic regression model (Fig. 3) (Table 4).

Figure 3 Non-linear relationship between NK cell count (2A) and NK cell proportion (2B) and 1-year relapse of MG.

The x-axis represents the NK cell count or proportion, while the y-axis indicates the risk of a 1-year relapse in patients with newly diagnosed MG. The middle line represents trend of 1-year relapse in NK cells. The upper and lower lines represent 95% confidence intervals.

Table 4 The nonlinearity addressing using two-piecewise linear model.

Exposure variables	Ln (NK cells count)	Ln (NK cells proportion)	
Fitting model using logistic regression model	0.49 (0.32, 0.75) 0.001	0.84 (0.57, 1.25) 0.3898	
Fitting model using two-piecewise linear model			
Inflection point	5.38	1.81	
<Inflection point	0.23 (0.11, 0.49) 0.0001	0.28 (0.06, 1.24) 0.0935	
>Inflection point	1.43 (0.62, 3.34) 0.4038	1.16 (0.65, 2.05) 0.6203	
P for log likely ratio test	0.009	0.114	
Notes.

We adjusted for Age; thymus status; Thymectomy; AChR-ab; MUSK-ab; Osserman classification; involvement of pharyngeal muscles; acetylcholinesterase inhibitor use; Immunosuppressants use

Discussion

To our knowledge, this is the first retrospective cohort study conducted at a single center to examine the relationship between the proportion and count of peripheral NK cells and 1-year recurrence risk in patients with MG. By fitting smooth curves and applying a two-piecewise linear model, we observed a nonlinear relationship (saturated effect) between Ln(PNKc) and the risk of recurrence. Particularly, within the range of 3.00–5.38, a higher level of Ln(PNKc) was associated with a decreased risk of recurrence. No significant correlations were observed beyond this range.

Although no epidemiological evidence exists that links NK cell count and proportion to the risk of MG recurrence, previous studies have confirmed several associations between NK cells and MG. In a study involving 33 patients with MG and 19 healthy controls, a reduction was observed in total NK cell count in the peripheral blood of patients with severe MG during acute exacerbation (Ge et al., 2023). Moreover, Nong et al. (2022) reported a significant reduction in the CD56+ NK cell count in patients with MG. These findings suggested a relationship between NK cell count and the onset and activity of MG; furthermore, they lay a theoretical foundation for utilizing NK cell count as a potential biomarker in MG prognosis. Moreover, the correlation between a higher baseline NK cell count and reduced recurrence risk can be explained by multiple mechanisms.

First, NK cells may identify and eliminate autoantibody-coated target cells. Clearing these antigen-presenting cells may alleviate pathogenic immune responses (Kiessling et al., 1975). Second, NK cells can secrete the immunosuppressive cytokine transforming growth factor-β. Elevated transforming growth factor-β levels can inhibit the activation of autoreactive T and B cells, thus reducing autoimmune reactions and aiding patients in maintaining a remission state (Chen et al., 2009; Sanjabi et al., 2017). Third, NK cells can directly eliminate autoreactive Tfh cells through their specific activating receptors (such as NKG2D) that recognize these cells, as well as their self-tolerant mechanisms and cytotoxic effects. This would diminish the supportive role of Tfh cells on B cells, reduce the production of pathogenic antibodies, and thereby decrease disease recurrence (Wang et al., 2017). Fourth, NK cells inhibit the differentiation of B cells into plasma cells, thereby decreasing the production of AChR antibodies and maintaining disease stability (Chien et al., 2011). Fifth, NK cells can exert negative regulatory effects on immune responses in the lymph nodes. An increase in the number of NK cells may enhance this negative regulatory effect, thus inhibiting autoimmune responses in the lymph nodes (Buentke et al., 2002). Sixth, when KIR on NK cell surfaces bind to HLA ligands, they send inhibitory signals to suppress the activation of autoreactive T cells. Moreover, an increased degree of KIR-HLA matching enhances this negative regulatory effect (Thananchai et al., 2007).

Hence, the correlation between an elevated NK cell count and improved prognosis in our study may be explained by a higher baseline NK cell count, indicating enhanced clearance function, which can be associated with a better prognosis. Notably, we observed a “saturation effect”, wherein an increase in the NK cell count beyond a certain range did not further reduce the risk of recurrence. Despite the constraints of observational studies preventing us from explaining this saturation effect, the discovery of this nonlinear relationship and the saturation effect was crucial for understanding the role of NK cells in autoimmune diseases and for developing optimal treatment strategies. This discovery revealed the complexity of NK cells in immune modulation, indicating that the effect of NK cells is not simply proportional and that the number of NK cells is not always directly linked to their immune regulatory effects. This finding highlights the limitations of current mechanisms and encourages further exploration of the functions and regulatory mechanisms of NK cells.

These findings are crucial for developing effective treatment strategies. This study identified a breakpoint between NK cell count and MG recurrence, demonstrating a nonlinear relationship between the two. In the lower range of values, a higher NK cell count was more beneficial for reducing MG recurrence, suggesting an immunoregulatory protective effect of NK cells on MG. However, above the upper range of values, the NK cell count was no longer linked to MG recurrence. Therefore, an optimal range of NK cell counts exists, which has significant implications for future MG treatment strategies. Maintaining NK cell counts within this range may reduce the recurrence risk and optimize treatment effects.

Our study has several strengths. First, the data used in our study were obtained from medical records from a hospital electronic medical record system. Thus, despite the potential flaws inherent to retrospective cohort studies, such as recall bias, the use of objective medical records helps alleviate the distortion of findings caused by such biases. Second, the recorders of these medical data were unaware that the data could be used in the future to analyze the association between NK cells and MG recurrence, naturally reducing the impact of observation bias on the results. Third, retrospective studies typically have missing data. However, we minimized the impact of missing data using various measures. Initially, we verified whether the absence of exposure variables influenced the results and determined that it did not bias our findings. In addition, we assessed the proportion of missing data and implemented suitable handling methods. Fourth, owing to the observational nature of the study, the results were inevitably influenced by confounding factors. Nevertheless, a series of rigorous adjustment strategies minimized the bias that these confounding factors might have introduced into our findings. Finally, we performed a series of rigorous sensitivity analyses to improve the robustness of the results.

This study has some limitations. First, our study was a single-center retrospective cohort investigation; hence, the generalizations of our findings to other contexts and populations should be approached with caution. Second, due to the observational nature of the study, we could only establish correlations and were unable to determine definitive causal relationships. Third, the study’s participant pool exclusively comprised patients with newly diagnosed MG; hence, our results may not be applicable to patients with a longer disease duration. Fourth, our adjustments accounted for measurable, but not unmeasurable, confounding factors. Fifth, despite our efforts to exclude infected patients based on exclusion criteria, we were unable to completely rule out the potential influence of underlying infections. Consequently, future studies will be based on prospective cohorts that will document and evaluate potential infection factors in greater detail, potentially with longitudinal follow-up, with the aim of gaining a deeper understanding of the dynamics of NK cell numbers. Sixth, this study focused exclusively on Chinese patients, which necessitates careful consideration while applying our findings to other ethnic groups. Nevertheless, we anticipate that similar studies in diverse populations will validate and enhance our results.Seventh, we used the same fluorescent marker of CD16 and CD56 to identify NK cells. However, this approach has some limitations. The most commonly used criterion for defining NK cells is CD3 negative and CD56 positive. Using the same fluorescent marker for CD16/CD56 can lead to misidentification of CD16+/CD56 cells as NK cells, which can affect the accuracy of the results. Although this method has been widely used in the past, it is not the best choice. We are aware that this labelling strategy may lead to misclassification of NK cells and thus affect the assessment of the immune status of MG patients. Therefore, we clearly point out this limitation in the discussion to improve the transparency and reproducibility of the study. Future studies should consider using a more stringent labelling strategy to ensure accurate identification of NK cells in order to better understand their role in the pathogenesis of MG. Finally, this study was a retrospective analysis based on clinical records from hospital diagnosis and treatment. As these data were collected during routine clinical practice rather than being designed for a specific research objective in a prospective study, the flow cytometry analysis did not include predefined subsets for NK cell subpopulations. This limitation in the study design restricted our ability to directly investigate the relationship between different NK cell subpopulations and the recurrence of various subtypes of MG. Future research should adopt a prospective study design with a comprehensive panel for NK cell subpopulations to deeply explore the role of different NK cell subsets in the recurrence of different MG subtypes. This will help to better understand the mechanisms of NK cells in MG recurrence and provide more precise guidance for clinical diagnosis and treatment.

Conclusions

In conclusion, NK cell count within a specific range was negatively correlated with the risk of MG relapse within a specific timeframe. This discovery supports the epidemiological evidence for a potentially critical role of NK cells in the development of MG. Future research can validate this association in large cohorts and aid in further exploration of the potential clinical application of NK cell counts and functional indicators in predicting MG relapse, which may form the basis for personalized treatment of this disease.

Supplemental Information

Data S1 Raw Data

Table S1 Antibody Information

Table S2 The description of missing data

Supplemental Information 4 Variables with a P value less than 0.1 associated with the outcome variable in univariate logistic regression

Table S4 Covariates impacting the association between Ln(NK cell count) and 1-year relapse-related risk (RR) by more than 10%

Table S5 Covariates impacting the association between Ln(NK cell proportion) and 1-year relapse-related risk (RR) by more than 10%

Supplemental Information 7 When Ln(PNKc) was within the range of 3.00-5.38, higher levels of Ln(PNKc) were associated with a lower risk of MG recurrence.

Additional Information and Declarations

Competing Interests

Author Contributions

Human Ethics

Data Availability

The authors declare there are no competing interests.

Yifan Zhang conceived and designed the experiments, performed the experiments, analyzed the data, prepared figures and/or tables, authored or reviewed drafts of the article, and approved the final draft.

Zhiguo Wen performed the experiments, prepared figures and/or tables, authored or reviewed drafts of the article, and approved the final draft.

Meiqiu Chen performed the experiments, prepared figures and/or tables, authored or reviewed drafts of the article, and approved the final draft.

Cong Xia conceived and designed the experiments, performed the experiments, analyzed the data, prepared figures and/or tables, authored or reviewed drafts of the article, and approved the final draft.

Fang Cai performed the experiments, prepared figures and/or tables, authored or reviewed drafts of the article, and approved the final draft.

Lan Chu conceived and designed the experiments, analyzed the data, prepared figures and/or tables, authored or reviewed drafts of the article, and approved the final draft.

The following information was supplied relating to ethical approvals (i.e., approving body and any reference numbers):

The Ethics Committee of the Affiliated Hospital of Guizhou Medical University approved the study (2022513).

The following information was supplied regarding data availability:

The raw measurements are available in the Supplementary Files.

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
