# Peer review of "Nonlinear relationship between circulating natural killer cell count and 1-year relapse rates in myasthenia gravis: a retrospective cohort study"

_PeerJ, doi:10.7717/peerj.18562_

## Round 0.1 · original submission · Major Revisions

Please correct the work according to the reviewer's requests

Reviewer 1 ·

Basic reporting

no comment

Experimental design

no comment

Validity of the findings

no comment

Additional comments

Generally, NK cells can be divided into CD56high and CD56dim subsets. The different NK subsets may play different role in autoimmune diseases. Further, the numbers of NK cells may change with aging. Myasthenia gravis (MG) clinically include many subtypes, such as early-onset MG, late-onset MG, thymoma assocated- MG and MUSK-IgG associated MG,etc. Zhang et al find a non-linear relationship between peripheral total NK cell count and MG recurrence in this manuscript. The study is interesting. However, considering the heterogeneity of NK cells and myasthenia gravis,it is better to analyze the role of different NK cell subsets in the recurrence of different MG subtypes.

Reviewer 2 ·

Basic reporting

the work is potentially interesting
However, numerous changes are needed

Experimental design

1. in the introductory part, add the classification of NK cells and the nature of NK cells as a special group of innate immunity cells: PMID: 31079327

2. in the material and methods section, clearly indicate the procedure for the analysis of NK cells and state it as previously recommended in works PMID: 27083212, PMID: 25933640

3. It is necessary to add the name of the flow cytometer, country of manufacture, type of device
4. It is necessary to add how many cells were examined for markers
5. it is necessary to present an image of a dot plot for patients in remission in relation to worsening of the disease, where differences in the phenotype of NK cells are clearly visible
6. in table 1, the values of NK are given as low, medium and high. None of the data has statistical significance due to this division. It would be best to reformulate the entire table and enter the original values and then compare them with the stage and all the parameters on the left side of the table. In this way, the findings for NK cells are divided into smaller groups and therefore have no significance, and the groups are divided by production. Try to do all the statistics by comparing the original values without grouping with the variables on the left.

Validity of the findings

it is necessary to perform new statistics

Additional comments

add representative pictures from flow

·

Basic reporting

"NK cells were defined as the cells that expressed CD3-CD56 surface markers" as that missing CD3 and expressing CD56

Authors suggest that NK cells may eliminate autoreactive T cells - how would they recognise them?

Article requires proofreading - typos, duplications of the begininngs of sentences etc.

Experimental design

How was the absolute number of NK cells counted?

Please list the antibodies used (fluorochrome, manufacturer, cat number or clone) and the manufacturer and model of flow cytometer.

Validity of the findings

Conclusions: first paragraph should be moved to discussion. 2nd paragraph repeats what is already discussed above. Real conclusions start at line 319 "In conclusion".

---

## Round 0.2 · Minor Revisions

Based on the reviewer's report, a final additional revision is needed

Reviewer 1 ·

Basic reporting

no

Experimental design

The total number of NK cells is influenced by multiple factors. Whether MG patients had concomitant infections when the blood samples were collected, especially in the group with higher NK cell numbers

Validity of the findings

the finding is some of intersesting and novelty although some limitation.

---

## Round 0.3 · Minor Revisions

Please address these final minor issues

·

Basic reporting

See below

Experimental design

See below

Validity of the findings

See below

Additional comments

1. Authors should provide enough data to warrant the repeatability of the study, thus the antibodies used have to be clearly listed. As authors used the ready mix, they should provide a supplementary table listing the antibodies in the mix (antigen, clone, fluorochrome). Antigen + fluorochrome is not enough to warrant repeatability. Additionally, authors should prepare gating strategy as a separate plot.

2. Use of CD16 and CD56 with the same fluorochrome is a limitation - the most widely used definition of NK cells is CD3 negative CD56 positive. If one uses CD16/CD56 with the same fluorochrome, it leads to enumeration of CD16+/CD56- cells as NK cells. It used to be a relatively wide spread approach, but it is not the best option. Thus authors should clearly discuss it as a limitation.

Since both are rather minor and straight forward I would suggest "minor revision".

---

## Round 0.4 · Minor Revisions

Thank you for submitting your paper for this journal

One of the reviewers, after reviewing your work, decided that the only thing remaining is to work on your gating strategy figure. This must be performed

·

Basic reporting

Since this is a 3rd review of that MS I will only focus on things that should be changed. The authors have acceptably answered my suggestions. The only problem now is the gating strategy figure. This should be prepared with dot plots - to clarify my point, I would suggest visiting this site: https://kcasbio.com/blogs/getting-a-go-to-gating-strategy/ - it is just one of the random results from a Google search. Please take a look at the figure. This is what the gating strategy should look like. From my perspective as soon as the authors prepare a proper gating strategy figure, the MS would be good to go.

Experimental design

as above

Validity of the findings

as above

Additional comments

as above

---

## Round 0.5 · accepted · Accept

The author accepted the reviewer's suggestions and added gate strategies in this revision.